# Comparison of Adenine-Induced Rat Models for Vascular Calcification in Chronic Kidney Disease

**DOI:** 10.3390/biology14070814

**Published:** 2025-07-04

**Authors:** Ho Won Kang, Ji Hye Kim, A Ro Yoon, Jahyung Kim, Joonhee Kim, Min Gyu Kyung, Dong Yeon Lee

**Affiliations:** 1Department of Orthopedic Surgery, Inha University Hospital, Incheon 22332, Republic of Korea; kangho-1@hanmail.net; 2Department of Orthopedic Surgery, Seoul National University College of Medicine, Seoul 03080, Republic of Korea; 3Department of Orthopedic Surgery, Seoul National University Hospital, Seoul 03080, Republic of Korea; 4Department of Orthopedic Surgery, Kyung Hee University Hospital at Gangdong, Seoul 05278, Republic of Korea

**Keywords:** adenine diet, chronic kidney disease, comparison, rat, vascular calcification

## Abstract

People with chronic kidney disease often suffer from a dangerous condition where calcium builds up in their blood vessels. This can lead to serious problems like heart disease or the need to amputate limbs. To find better treatments, scientists need reliable ways to study this condition. In this study, we tested different ways to feed rats a substance called adenine, which can damage the kidneys and mimic the disease seen in humans. We tried various amounts and feeding periods to see which method would cause calcium to build up in the blood vessels, while still keeping the animals healthy enough to study. We found that giving rats a higher dose for 12 weeks, followed by a lower dose for 6 weeks, led to a strong calcium buildup with fewer deaths. This approach gives researchers a new and effective way to study blood vessel disease linked to kidney problems. Understanding this process in animals can help scientists develop better treatments and prevention strategies for people in the future.

## 1. Introduction

Vascular calcification is the ectopic deposition of calcium-phosphate complexes in blood vessels and is a major risk factor for cardiovascular mortality in patients with chronic kidney disease (CKD) and diabetes [1,2,3]. It also occurs in peripheral vessels, leading to peripheral artery disease (PAD), which results in symptoms such as ischemic pain, claudication, and ischemic ulcers [4]. PAD can lead to debilitating conditions, such as critical limb ischemia (CLI), which may necessitate amputation [5]. PAD prevalence is elevated in patients with CKD by up to 46%, and CLI is associated with an annual amputation rate exceeding 25% [6,7].

Vascular calcification occurs in both the intima and media layers of the arterial wall. Intimal layer calcification is closely related to lipid deposits, causing obstructive arterial disease [3,8]. The medial layer calcification contributes to vascular stiffening and directly affects PAD and CLI [4]. Currently, the primary treatment method for CLI involves the amputation of necrotic toes or feet. Major amputation results in a 5-year mortality rate ranging from 53% to 100% [9]. Furthermore, approximately one-third of limb amputees suffered from clinically significant depression [10].

Despite improvements in revascularization surgery, earlier assessment at specialist high-risk foot clinics, and better management of the early stages of diabetes, partial foot amputation incidence continues to rise and is projected to more than triple by 2050 [11]. Even after partial foot amputation, poor wound healing and frequent re-amputations are common, highlighting the urgent need for treatments that target vascular calcification directly [12]. However, no treatment is currently available for medial vascular calcification, making the development of relevant animal models essential.

Animal models of medial vascular calcification predominantly utilize rats induced with CKD. Rat models of CKD induced by partial nephrectomy or adenine-rich diet supplementation are often used to study vascular calcification [13]. Because the partial nephrectomy model requires a surgical procedure, the adenine-induced rat model has become widely adopted [14]. Supplementation with a 0.75% adenine diet for 4 wk has gained general acceptance as a model for studying kidney damage [13,15]. Excess dietary adenine is oxidized to 2,8-dihydroxyadenine and excreted by the kidney, precipitating in the tubules and resulting in tubular injury [16,17].

However, dietary protocols have not yet been standardized [18,19]. The commonly used 4 wk 0.75% adenine protocol corresponds to approximately 2 y in human lifespan [20]. It is unsuitable for mimicking the slowly progressing nature of human CKD. Furthermore, the 0.75% adenine diet rat model demonstrates a marked decline in survival rate after the fourth week, reaching only 12.5% at 8 wk [21]. Therefore, various adenine concentrations (0.075%, 0.25%, 0.5%, and 0.75%) have been used to induce kidney damage [19]. Diwan et al. found that 0.25% adenine causes chronic kidney damage with functional and structural changes after 16 wk, similar to human CKD [13]. However, as calcification was not identified in that study, it is not suitable as an animal model of vascular calcification. Because vascular calcification is a key contributor to CLI-related amputations in patients with CKD, inducing both CKD and vascular calcification is essential for research on CLI treatment.

To the best of our knowledge, no study has compared a long-term animal model of vascular calcification that mimics human CKD. This study aimed to determine the optimal concentration and duration of an adenine diet in inducing vascular calcification in a rat model and to determine their survival and calcification rates.

## 2. Methods

### 2.1. Experimental Animals

Male Sprague–Dawley rats (Koatech, Pyeongtaek, Gyeonggi-do, Republic of Korea) weighing 350–400 g were divided into six groups after a 1 wk acclimatization period. Two rats per cage were accommodated in a temperature- and humidity-controlled environment (18–20 °C at 30–70% humidity) with a 13–14-h light-dark cycle (130–325 lux). Standard and adenine diets were purchased from DooYeol Biotech (Seocho-gu, Seoul, Republic of Korea). Food and water were placed on the side of the rat’s cage, allowing for free access throughout the study period. The experimental protocols were approved by the Institutional Animal Care and Use Committee of Seoul National University Hospital (IACUC No. 24-0037-S1A0).

### 2.2. Experimental Protocols

The control group (n = 12) received a standard diet for 18 wk; group 1 (n = 12), a 0.5% adenine diet for 4 wk, followed by a standard diet for 14 wk; and group 2 (n = 12), a 0.5% adenine diet for 4 wk, followed by a 0.25% adenine diet for 14 wk. Group 3 comprised 70 rats provided with a 0.5% adenine diet for the first 12 wk. In rats that survived for up to 12 wk, those in group 3A (n = 12) received a standard diet, and those in group 3B (n = 12) received a 0.25% adenine diet after 12 wk. Group 3C (n = 12) was continuously fed a 0.5% adenine diet for 18 wk (Figure 1).

Body weights were checked at 0, 4, 12, and 18 wk. Serum biochemistry was analyzed at 18 wk. Rats were anesthetized, and blood was collected. The serum was stored at −20 °C until further analysis. The biochemistry was analyzed at the Biomedical Research Institute of Seoul National University Hospital, using a Hitachi 7180 auto analyzer (Hitachi High Technologies Co., Tokyo, Japan). The following biochemistry were performed: serum calcium (mg/dL), serum phosphorus (mg/dL), serum glucose (mg/dL), serum BUN (mg/dL), serum creatinine (mg/dL), serum uric acid (mg/dL), serum sodium (mmol/L), serum potassium (mmol/L), and serum chloride (mmol/L).

In the control group and group 1, four live rats were sacrificed for a histological examination at 4, 12, and 18 wk, respectively. In group 2, four rats were sacrificed at 12 wk, and eight rats were sacrificed at 18 wk. In group 3, six rats were sacrificed at 8 wk for a histological examination, and 10 rats were sacrificed at 12 wk. At 18 wk, all surviving rats were sacrificed. The rats were sacrificed by sevoflurane anesthesia, and the thoracic and abdominal aortas were collected for histological examination.

### 2.3. Histology

For each animal, the harvested aorta was processed into one to three tissue sections prior to embedding for von Kossa staining. 5 µm sections of the paraffin-embedded aortas were processed for von Kossa staining. Sections were deparaffinized and rehydrated in distilled water. Subsequently, the cells were treated with 1% silver nitrate (*w*/*v*) and exposed to ultraviolet light for 20 min. Next, the sections were placed in 5% sodium thiosulfate (*w*/*v*) for 5 min and counterstained with nuclear fast red for 5 min.

The degree and prevalence of vascular calcification were semi-quantitatively assessed using a four-grade scale based on previous studies: no calcification (no visible calcified area), mild calcification (1–33% of the aortic ring affected), moderate calcification (34–66%), and severe calcification (67–100%) [21].

### 2.4. Statistical Analysis

The statistical analyses were performed using the Statistical Package for Social Science, version 25 (IBM Corp., Armonk, NY, USA). The quantified data were presented as mean ± SD. The Kaplan–Meier method and log-rank test were used to compare survival rates between groups. Differences between groups were analyzed using a one-way analysis of variance followed by Scheffe’s post hoc test. Statistical significance was set at a 95% confidence level (*p* < 0.05).

## 3. Results

### 3.1. Body Weight

The rats’ body weight averaged 382.7 ± 24.1 g after the 1 wk acclimatization period. In the control group, the average body weight increased continuously to 435.9 ± 22.1 g at 4 wk, 491.0 ± 32.1 g at 12 wk, and 513.9 ± 37.1 g at 18 wk. In the groups given a 0.5% adenine diet for 4 wk, the body weight decreased to 317.9 ± 27.4 g at 4 wk. In group 1, which received a standard diet after 4 wk, the body weight increased to 492.9 ± 30.7 g at 12 wk and 545.6 ± 12.9 g at 18 wk. In group 2, which received the 0.25% adenine diet after 4 wk, the body weight increased to 366.3 ± 32.1 g at 12 wk and 384.6 ± 32.2 g at 18 wk. In group 3, which received a continuous 0.5% adenine diet for 18 wk, the body weight decreased to 268.9 ± 39.7 g at 12 wk. For groups 3A and 3B, who received 0.5% adenine for the first 12 wk, followed by a standard diet for group 3A and a 0.25% adenine diet for group 3B, the body weight increased to 397.0 ± 42.3 g and 319.8 ± 38.5 g at 18 wk for the respective groups. Group 3C, who received a 0.5% adenine diet throughout the study, had a decrease in body weight to 245.9 ± 50.2 g at 18 wk (Figure 2) (Appendix A).

### 3.2. Serum Biochemistry

Serum calcium levels did not show significant differences among the groups at 18 wk, while serum phosphorus levels showed a significant difference between group 3C and the control group. Serum blood urea nitrogen (BUN) levels in groups 2, 3A, 3B, and 3C differed significantly from those in the control group. Serum BUN levels in groups 2, 3B, and 3C differed significantly from those in Group 1. Serum BUN levels in group 3C differed significantly from those in group 2. Serum creatinine levels in groups 2, 3A, 3B, and 3C differed significantly from those in the control group and group 1. Serum uric acid levels in group 3B differed significantly between the control group and group 1, whereas serum uric acid levels in group 3C differed significantly from those in the control group and in groups 1 and 2. Serum potassium levels in groups 3A, 3B, and 3C differed significantly from the control group and group 1, with group 3B also differing significantly from group 1. The serum chloride level in group 3B differed significantly from the control group and groups 1 and 2, while group 3C differed significantly from the control group and groups 1, 2, and 3A. Serum glucose and sodium levels did not differ significantly. No parameter differed significantly between groups 3B and 3C (Figure 3) (Appendix A).

### 3.3. Survival Rate and Calcification Rate (Table 1 and Table 2)

In the control group and group 1, no calcification was observed histologically, and none of the rats died until the 18th week. In group 2, all of the rats survived until the 18th week, and mild calcification was observed in one of the eight specimens in the 18th week. However, no calcification was observed at 12 wk.

In group 3, mortality occurred from 7 wk onward. One rat died at 7 wk, and two died at 8 wk. Since then, more rats continued to die, and 15 rats died up until the 12th week. Histology was examined in six sacrificed rats and one of the dead rats at 8 wk. Calcification was observed in only one dead rat out of a total of seven specimens (14.3%). At 12 wk, 6 dead rats and 10 sacrificed rats were examined histologically. Calcification was observed in 3 dead rats and 2 sacrificed rats out of 16 rats (31.3%). After 12 wk, the 36 rats that were not sacrificed or had died were evenly divided into groups 3A, 3B, and 3C. In group 3A, six out of 12 rats survived until the 18th week. From the histological results of the two dead rats and six surviving rats, severe calcification was only observed in one surviving rat (12.5%). Of the 12 rats in group 3B, 9 survived, and upon histological examination of these rats, calcification was confirmed in 6 (1 mild, 1 moderate, and 4 severe calcification) (66.7%). In group 3C, 6 of the 12 rats survived, and histological analysis was conducted on these rats. Calcification was observed in four of the six surviving rats (one moderate and three severe calcifications) (66.7%) (Figure 4).

The calcification rates in groups 3B and 3C were similar, but the survival rates after 12 wk were 75.0% and 50.0%, respectively. However, the survival rates between the two groups did not differ significantly (*p* = 0.148) (Figure 5).

**Table 1 biology-14-00814-t001:** Survival rate.

	Survival Rate (%)			
Group	4 wk	8 wk	12 wk	18 wk
Control (n = 12)	12/12 (100)	8/8 (100)	8/8 (100)	4/4 (100)
Group 1 (n = 12)	12/12 (100)	8/8 (100)	8/8 (100)	4/4 (100)
Group 2 (n = 12)	12/12 (100)	12/12 (100)	12/12 (100)	8/8 (100)
Group 3 (n = 70)	70/70 (100)	67/70 (95.7)	46/64 (71.9)	
Group 3A (n = 12)				6/12 (50.0) *
Group 3B (n = 12)				9/12 (75.0) *
Group 3C (n = 12)				6/12 (50.0) *

* Survival rate after 12 wk.

**Table 2 biology-14-00814-t002:** Calcification rate.

	Calcification Rate (%)			
Group	4 wk	8 wk	12 wk	18 wk
Control (n = 12)	0/4 (0.0)	-	0/4 (0.0)	0/4 (0.0)
Group 1 (n = 12)	0/4 (0.0)	-	0/4 (0.0)	0/4 (0.0)
Group 2 (n = 12)	-	-	0/4 (0.0)	1/8 (12.5)
Group 3 (n = 70)	-	1/7 (14.3)	5/16 (31.3)	
Group 3A (n = 12)				1/8 (12.5)
Group 3B (n = 12)				6/9 (66.7)
Group 3C (n = 12)				4/6 (66.7)

## 4. Discussion

In this study, it was observed that the administration of 0.5% adenine for less than 4 wk did not induce vascular calcification, while administration for more than 8 wk did. However, the mortality rate began to increase after 8 wk, and the survival rate was 71.9%, with a calcification rate of 31.3% in the 12th week. Upon changing to a standard diet after 12 weeks, the calcification rate did not increase further by the 18th week, remaining at 12.5%. Changing to a 0.25% adenine diet or continuing a 0.5% adenine diet resulted in a calcification rate of 66.7%. However, in terms of survival rate, the 0.25% adenine diet resulted in a higher calcification rate of 75.0% compared to 50.0% with the continuous 0.5% adenine diet. Therefore, a rat model with a 0.5% adenine diet for 12 wk, followed by a 0.25% adenine diet for 18 wk, is optimal in preparing an animal model for vascular medial calcification research.

Rodent models of CKD are typically established using either surgical approaches or nephrotoxic dietary interventions. Among surgical methods, 5/6 nephrectomy is the most commonly employed technique [18]. The surgical removal of one kidney and ablation of a portion of the remaining kidney impairs renal function [13]. However, surgical methods have drawbacks, such as high mortality rates, low consistency of surgical outcomes, and the necessity for surgical procedures [22]. Furthermore, it does not mimic the gradual progression of CKD in humans because renal function deteriorates abruptly after surgery.

The adenine-containing diet is a widely utilized method for inducing nephrotoxicity in experimental models of CKD [23]. Administering an adenine diet to rats induces functional changes similar to those observed in human CKD, leading to the rapid onset of renal damage and subsequent vascular calcification. Adenine is a purine nucleobase that forms part of DNA and plays various roles in biochemical and physiological processes within cells [24]. Adenine is a common dietary component present in seafood, beer, and organ meats [25]. It is efficiently salvaged by the normal adenine phosphoribosyl transferase (APRT) pathway. However, when adenine is administered in excess, APRT activity is saturated, and adenine is oxidized by xanthine dehydrogenase to 2,8-dihydroxyadenine, which forms precipitates and crystals in the renal tubules, resulting in tubular fibrosis, inflammation, and obstruction [17]. To enhance CKD induction and accelerate vascular calcification, phosphate, calcium, calcitriol, and vitamin D are often administered in combination with an adenine diet [26,27,28].

CLI is caused by the calcification of the medial layer of the arteries, which leads to amputation in patients with CKD. Therefore, it is necessary to induce not only CKD but also vascular calcification for research on the treatment of CLI. Vascular calcification and subsequent CLI, which occur in the late stages of CKD, progress gradually over several years in humans, necessitating the establishment of long-term animal models. The commonly used 0.75% adenine diet rat model leads to rapid kidney damage, making it difficult to administer for more than 4 wk. In one study, all of the mice died within 6 d of receiving the 0.75% adenine diet [29]. Diwan et al. have reported a long-term model mimicking human CKD by administering 0.25% adenine to rats for 16 wk [19]. However, they did not find any vascular calcification. In this study, the 0.25% adenine diet was ineffective at inducing vascular calcification.

This study’s findings indicate that while rats on a standard diet continued to gain weight, those on a 0.5% adenine diet experienced a continuous decrease in weight. Following 4 wk on a 0.5% adenine diet, if the rats were then switched to a standard diet, their weight began to increase. By the 12th week, their weight recovered to levels comparable to those of rats on a standard diet only. Even after 12 wk on a 0.5% adenine diet, changing to a standard diet led to a return to weight gain. After inducing kidney damage with a 0.5% adenine diet for 12 wk, group 3A then received a standard diet. At 18 wk, serum biochemistry findings indicated that group 3A had higher chloride levels compared with groups 3B and 3C. Furthermore, group 3A had uric acid levels that, unlike groups 3B and 3C, did not differ significantly from those in the control group. These observations suggest that a standard diet can help kidney function recovery.

However, changing from a 0.5% adenine diet to a 0.25% adenine diet resulted in a maintenance of body weight. At 18 wk, BUN, creatinine, and uric acid levels in group 3B were all lower than those in group 3C, but this difference was not statistically significant. Therefore, while the 0.5% adenine diet rapidly induced kidney damage, a 0.25% adenine diet either gradually induced kidney damage or maintained the level of damage in the already affected kidneys.

Since group 3A received a standard diet after 12 wk, it was expected that the survival rate would be higher than that in group 3B; however, group 3A had a 50% survival rate, which was lower than that in group 3B (75%). While efforts were made to control the breeding environment equally, the possibility that the physical conditions of the purchased rats or the breeding environment may have differed slightly was considered, which affected the rats, as not all rats could be experimented on at the same time. Furthermore, two rats were accommodated in one cage, and whenever a rat in the same cage died, some rats did not eat even a standard diet, and their condition deteriorated in group 3A. The exact cause for this is unknown; however, it is clear that 0.25% adenine does not increase mortality compared to the standard diet, considering that no rat in group 2 died during the study period.

This study had several limitations. First, histological examination was performed on the aortas despite the fact that CLI is caused by the calcification of peripheral arteries. Second, vascular calcification was evaluated semi-quantitatively by histological analysis, without precise measurement of its extent or quantity. Third, the stages before vascular calcification detection via von Kossa staining were not assessed. This study is nonetheless relevant, considering that CLI occurs significantly after advanced vascular calcification. Fourth, since the rats had free access to food, it was challenging to determine whether their condition changed due to the adenine diet or because they were starving. Fifth, not all biopsies were performed on dead rats. In the case of dead rats, the biopsy was difficult because the specimen was in poor condition. Sixth, the rat model with 0.5% adenine for 12 wk followed by 0.25% adenine until 18 wk did not show a statistically significant difference in survival rates compared to rats on a continuous 0.5% adenine diet for 18 wk. However, there was a difference between the nine of 12 rats in group 3B and six of 12 rats in group 3C. The small sample size likely contributed to the lack of statistically significant results. Seventh, only male rats were used in this study. The adenine model effectively induces CKD in both sexes, but male rats are more susceptible to renal injury. Therefore, male rats were selected for this experiment [24,30,31].

## 5. Conclusions

In this study, we proposed a rat model with a 0.5% adenine diet for 12 wk, followed by a 0.25% adenine diet for up to 18 wk, as an optimal model for vascular medial calcification that mimics human CKD. Further research is needed to improve the survival rates.

## Figures and Tables

**Figure 1 biology-14-00814-f001:**
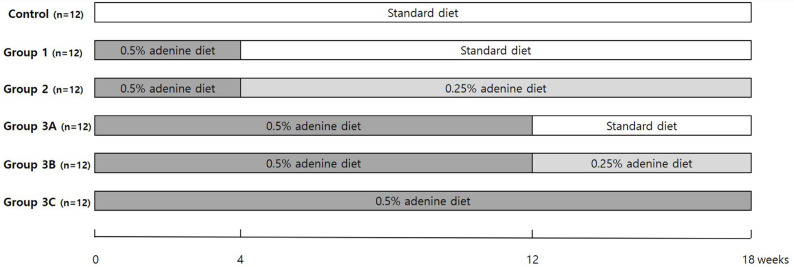
Experimental groups.

**Figure 2 biology-14-00814-f002:**
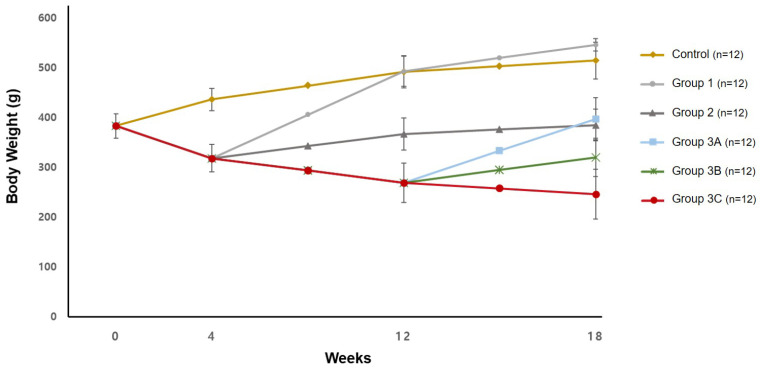
The changes in body weight for each group.

**Figure 3 biology-14-00814-f003:**
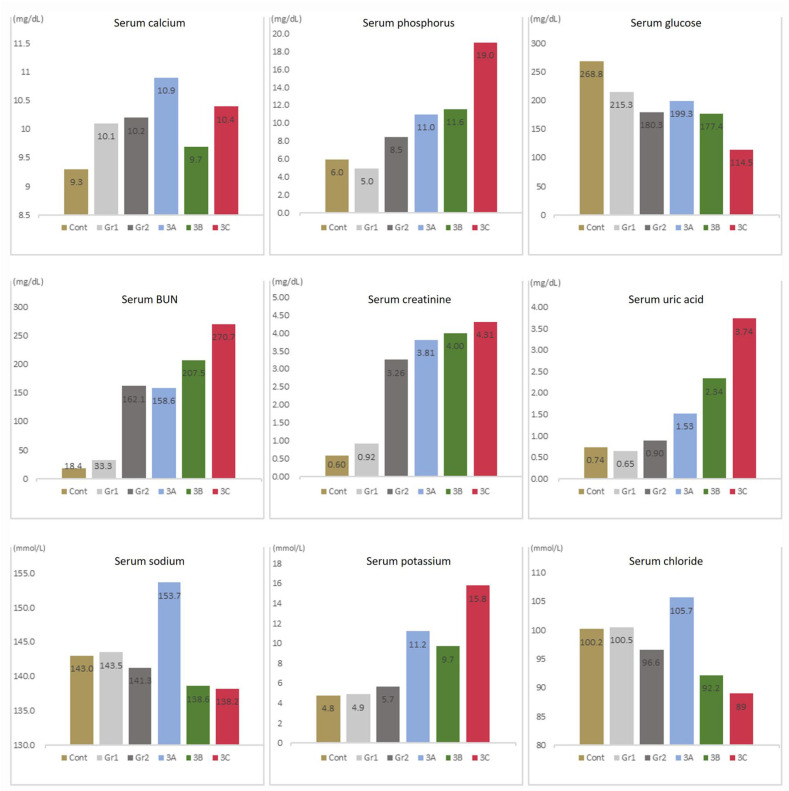
Serum biochemistry results at 18 wk by group. Blood urea nitrogen (BUN).

**Figure 4 biology-14-00814-f004:**
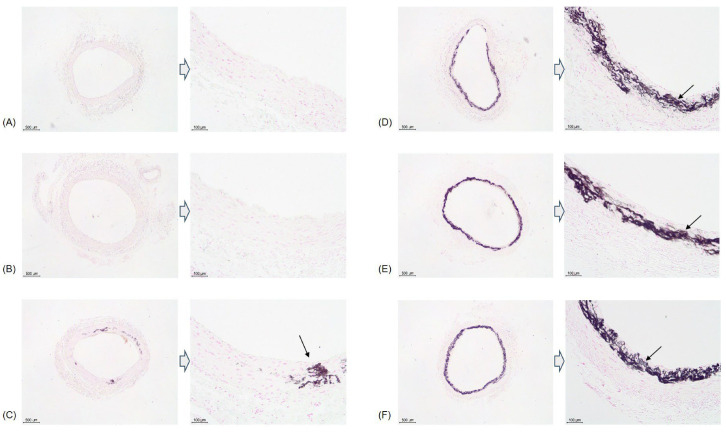
Examples of vascular calcification observed at 18 wk using von Kossa staining. No calcification was observed in the control group (**A**) and group 1 (**B**). Group 2 (**C**) showed mild calcification in the medial layer of the aorta. Groups 3A (**D**), 3B (**E**), and 3C (**F**) showed severe calcification. Arrows indicate vascular calcification.

**Figure 5 biology-14-00814-f005:**
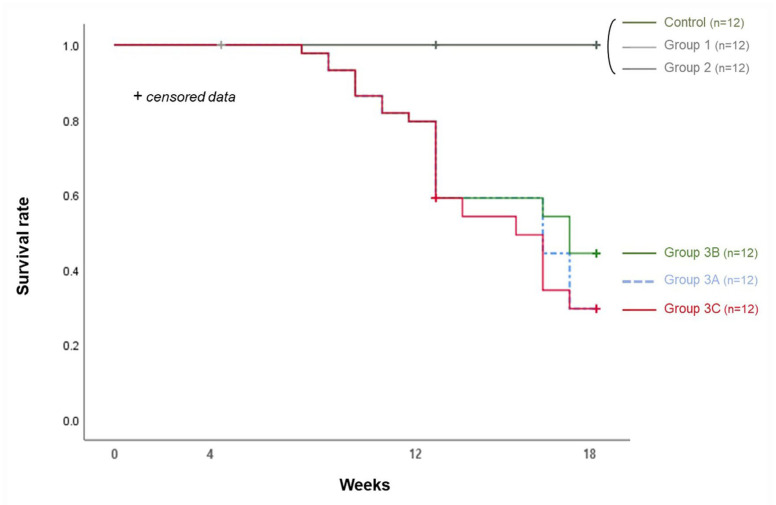
Survival rates of each group for 18 wk.

## Data Availability

All data are contained within the article or Appendix A.

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
