# Peer review of "Comparison of Adenine-Induced Rat Models for Vascular Calcification in Chronic Kidney Disease"

_biology, 2025, doi:10.3390/biology14070814_

Round 1
Reviewer 1 Report
Comments and Suggestions for Authors
This manuscript provides a comparison of models for adenine-induced calcification in rats, with different concentrations and durations of the adenine diets compared. Based on their findings, the authors conclude that 0.5% adenine for 12 weeks followed by 0.25% adenine for 6 weeks, effectively induced vascular calcification while maintaining acceptable survival rates in the animals. The manuscript is well written, and the findings are novel and of interest to the field.
Please address the following comments:
- Lines 117-118: (w/v) or (v/v) was not provided for solutions made as a %.
- Line 115: I presume '5ml sections' should read '5 µm sections' - please clarify?
- Histology: How many tissue sections were analysed with von kossa staining for each animal? Can you provide evidence of how consistent the calcification was throughout the entirety of the aorta?
- All figure legends should provide n numbers.
- Serum biochemistry measurements are provided in Table 1 - however, information for how this analysis was carried out should be provided in the Methods.
- Table 1 - the readings for each measurement provided in Table 1 are not clear to read. As there does seem to be large variability in some of the measurements (e.g. BUN), please can you provide these data in graphs instead.
- Scale bars should be provided in Figure 3.
- In the results section, you comment on the number of animals in which calcification was detected. However, was the extent of calcification in each animal consistent e.g. for rats on the 0.5% adenine diet for 18 weeks, was extensive calcification throughout the medial layer detected in all of the 4/6 animals? How many sections from each aorta did you analyse?
- Survival curves should be provided for all groups of rats.
- In this study, the authors used male rats – but sex bias still remains a considerable problem for animal studies. Have the authors examined this adenine model in female rats as a comparison?
Author Response
- Lines 117-118: (w/v) or (v/v) was not provided for solutions made as a %.
; Thank you for pointing this out. We have stated this in lines 128–129 of the revised manuscript as follows:
‘1% silver nitrate(w/v)’, ’5% sodium thiosulfate (w/v)’
- Line 115: I presume '5ml sections' should read '5 µm sections' - please clarify?
; Thank you for pointing this out. We have revised this in lines 126 as follows:
'5 µm sections'
- Histology: How many tissue sections were analysed with von kossa staining for each animal? Can you provide evidence of how consistent the calcification was throughout the entirety of the aorta?
; Thank you for pointing this out. We examined 1–3 sections stained with von Kossa from each aorta, and this information has added in line 125-126. However, since we did not assess the extent of calcification across the entire aorta, we have acknowledged this as a limitation in the manuscript (line 289). Additionally, to provide a more detailed assessment, we classified the degree of calcification observed in a single section as mild, moderate, or severe (lines 131–132, 194, 196, and 199).
- All figure legends should provide n numbers.
; Thank you for pointing this out. We have added the n numbers to all figures.
- Serum biochemistry measurements are provided in Table 1 - however, information for how this analysis was carried out should be provided in the Methods.
; Thank you for pointing this out. We have added the information of biochemistry measurements in line 109-116 as follows:
“ Serum biochemistry was analyzed at 18 wk. Rats were anesthetized, and blood was collected. The serum was stored at −20 °C until further analysis. The biochemistry was analyzed at the Biomedical Research Institute of Seoul National University Hospital, using a Hitachi 7180 auto analyzer (Hitachi high technologies Co., Tokyo, Japan). The following biochemistry were performed: serum calcium (mg/dL), serum phosphorus (mg/dL), serum glucose (mg/dL), serum BUN (mg/dL), serum creatinine (mg/dL), se-rum uric acid (mg/dL), serum sodium (mmol/L), serum potassium (mmol/L), and serum chloride (mmol/L).”
- Table 1 - the readings for each measurement provided in Table 1 are not clear to read. As there does seem to be large variability in some of the measurements (e.g. BUN), please can you provide these data in graphs instead.
; Thank you for pointing this out. We have converted Table 1 into a new figure (Figure 3) and added it to the manuscript. The original Table 1 has been changed to Supplementary Table 1.
- Scale bars should be provided in Figure 3.
; Thank you for pointing this out. We have added scale bars to the figure (revised manuscript Figure 4) and, following the editor’s recommendation, included representative samples from all groups.
- In the results section, you comment on the number of animals in which calcification was detected. However, was the extent of calcification in each animal consistent e.g. for rats on the 0.5% adenine diet for 18 weeks, was extensive calcification throughout the medial layer detected in all of the 4/6 animals? How many sections from each aorta did you analyse?
; Thank you for pointing this out. We examined 1–3 sections stained with von Kossa from each aorta, and this information has added in line 125-126. However, since we did not assess the extent of calcification across the entire aorta, we have acknowledged this as a limitation in the manuscript (line 289). Additionally, to provide a more detailed assessment, we classified the degree of calcification observed in a single section as mild, moderate, or severe (lines 131–132, 194, 196, and 199).
- Survival curves should be provided for all groups of rats.
; Thank you for pointing this out. We have added survival curves for all groups in figure 5.
- In this study, the authors used male rats – but sex bias still remains a considerable problem for animal studies. Have the authors examined this adenine model in female rats as a comparison?
; Thank you for pointing this out. I agree with this comment. The adenine model effectively induces CKD in both sexes, but male rats are more susceptible to renal injury. Therefore, male rats were selected for this experiment. This has been added to the limitation section with references (line 301-303).
Reviewer 2 Report
Comments and Suggestions for Authors
The manuscript “Comparison of Adenine-Induced Rat Models for Vascular Calcification in Chronic Kidney Disease” presents an interesting study addressing the development of a suitable model to evaluate vascular calcification in the context of chronic kidney disease. I would like to share a few comments and suggestions:
-
In Figure 2, it would be helpful to indicate statistically significant differences directly on the lines where applicable.
-
Table 1 could be improved and reformatted for better clarity and presentation.
-
The authors should specify whether they used Kaplan-Meier curves to analyze survival data.
-
Additionally, in the Results section under Calcification and Survival, the number of rats and the time points for histological analysis are already described in the experimental design section — it may be unnecessary to repeat this information.
-
In Figure 3, it would be useful to mark or indicate the calcified areas directly on the images.
-
On line 240, there appears to be a symbol (자) that should be removed.
Author Response
- In Figure 2, it would be helpful to indicate statistically significant differences directly on the lines where applicable.
; Thank you for pointing this out. I agree with this comment. I felt that indicating all statistical differences directly in the figure would make it overly complex. Therefore, we presented the mean and standard deviation of body weights, and statistically significant differences in Supplementary Table 1.
- Table 1 could be improved and reformatted for better clarity and presentation.
; Thank you for pointing this out. I agree with this comment. In response to the other reviewer’s suggestion to present Table 1 as a figure, we have converted Table 1 into a figure (now Figure 3). The original contents of Table 1, with the addition of body weight data, are now presented as Supplementary Table 1.
- The authors should specify whether they used Kaplan-Meier curves to analyze survival data.
; Thank you for pointing this out. I used the Kaplan-Meier method , as noted in line 138 of the revised manuscript.
- Additionally, in the Results section under Calcification and Survival, the number of rats and the time points for histological analysis are already described in the experimental design section — it may be unnecessary to repeat this information.
; Thank you for pointing this out. As suggested, we have deleted the relevant section from the manuscript.
- In Figure 3, it would be useful to mark or indicate the calcified areas directly on the images.
; Thank you for pointing this out. As suggested, we have deleted the relevant section from the manuscript. As suggested, we have added arrows to indicate the calcified areas in the figure (revised manuscript figure 4).
- On line 240, there appears to be a symbol (자) that should be removed.
; Thank you for pointing this out. We have replaced the character “자” with “wk” in the revised manuscript.